# Production Technologies and Provenance of Ceramic Materials from the Earliest Foundry of Pre-Roman Padua, NE Italy

Elena Mercedes Pérez-Monserrat [1,2,*], Lara Maritan [2], Vanessa Baratella [1] and Massimo Vidale [1]

1 Department of Cultural Heritage, University of Padua, 35139 Padua, Italy; vanessa.baratella@unipd.it (V.B.); massimo.vidale@unipd.it (M.V.)
2 Department of Geosciences, University of Padua, 35131 Padua, Italy; lara.maritan@unipd.it
* Correspondence: elenamercedes.perezmonserrat@unipd.it

**Abstract:** The earliest foundry site of Pre-Roman Padua, in the Veneto region (north-eastern Italy), provided evidence of craft activities and residential areas dated between the end of the 9th and 1st centuries BCE. Common and fine wares were found, most of them belonging to two main ceramic classes: (i) highly dark-colored bodies with silicate inclusions; (ii) carbonate-tempered pots. Macroscopic and petrographic (POM) descriptions and spectroscopy—mineralogical (XRPD) and geochemical (XRF)—analyses were carried out to define the provenance and technological choices (production recipes and firing conditions). The first ceramic class comprised diverse local/regional productions made with a common geo-resource, consisting in quartz-rich illitic clays, for which pit firing conditions generally under 850–900 °C were adopted. The purification of the starting base clays, the tempering of the clayey pastes with silica-rich grains and/or the polishing of surfaces were accomplished for fine wares. The carbonate-tempered pottery probably corresponded to an allochthonous ceramic class that might be used in the trade of specific contents. As such pots were made out of different illitic clays and under different redox conditions, they might correspond to diverse productions. The adoption of precise choices in very specific wares, not belonging to any of these two main ceramic classes, suggests the trade of fine wares in the Veneto during the First Iron Age.

**Keywords:** lithic markers of provenance; clay purification; silicate and carbonate tempering; regional and extra-regional pottery exchange

## 1. Introduction

Archaeometry studies based on morphological, macro/microscopic, textural and compositional analysis provide data about the manufacturing of ancient ceramics, valuable information when documental sources are lacking. Macroscopically, specific treatments adopted during the production can be recognized and, from the color of the ceramics, primary evidence about the firing conditions can be collected [1–4]. The petrographic description allows the identification of production recipes and the solving of provenance issues, especially for coarse wares in which mineral markers are present [5]. Mineral assemblages permit the constraining of the firing temperature and/or the alteration processes experienced by the sherds, while chemical analyses are used to define the provenance of fine ware [5–8].

As pottery shapes artefacts that perform specific functions, precise technological choices should be adopted. For instance, the use of clayey materials with a certain texture and composition or the adoption of close/open firing methods, issues that can be defined by archaeometry analysis [9,10]. For practical and aesthetical reasons, the surfaces of the pots are frequently polished. The polishing is normally driven by pression, enhancing the higher concentration of small particles on the surface, which has a glossy appearance after firing. This process fosters sintering and, therefore, the achievement of low porosity and high protection on surfaces [11,12].

Procedures such as the purification of the starting raw clays or the tempering of the clay pastes are often recognized in ancient wares, especially by textural analysis [13]. The

production of fine wares required very fine-grained or depurated clayey materials. As the small particle sizes are selected/concentrated, fine wall shapes can be modeled and the sintering is likewise enhanced [11]. The properties of ceramic products can be improved by adding to clay pastes non-plastic materials known as temper, such as silicate and carbonate inclusions or crushed fired pottery, i.e., grog [13,14]. For instance, the addition of quartz diminishes the plasticity of clays [15] and carbonates act as a flux for getting a lower sintering temperature [16].

Although carbonate addition improves the workability of the ceramic pastes and the hardness of pottery [17,18], the presence of calcite may provoke cracks when carbonate inclusions occur in large grains and the clay paste is fired at a temperature exceeding that of calcite decomposition (above 800–850 °C). This defect, known as "lime-blowing", is due to the increase in volume caused by the hydration of the lime still available after firing (not reacted with the surrounding groundmass) [19,20]. The hydrated lime (portlandite) is very unstable and reacts with the atmospheric $CO_2$, forming secondary calcite that may induce the cracking of the ceramic bodies. The harmfulness of calcite in ceramics depends to a large extent on the grain size and on the content of clay minerals in the starting material [21]. Inclusions, originally occurring in the base clays or deliberately added as temper, may entail optimal markers of provenance of the areas where the used clayey materials outcropped and/or the wares were produced [5]. Hence, their identification is essential to resolve the provenance question and to state possible commercial trades [22].

In the present work, a multi-analytical study based on macroscopic, petrographic, mineralogical and geochemical analyses was conducted on potsherds found in the earliest foundry site of Pre-Roman Padua (Veneto region, north-eastern Italy). The main objective was to answer archaeological queries about provenance and production technology, with the aim also of defining possible regional and extra-regional pottery exchanges.

## 1.1. Rock and Mineral Fragments and Clay Fractions of Padua's Territory Sediments

The area of Padua is located on the eastern side of the Po plain and it lies on the quaternary alluvial sediments of the Adige, Brenta and Bacchiglione rivers (Figure 1a). The deposits transported by the Adige are chiefly characterized by the presence of fragments of metamorphic (quartzite and mica–schist) and acid volcanic (rhyolite and trachyte) rocks, while the Brenta and Bacchiglione rivers mainly transport fragments of phyllites, granitoids, limestones, dolomites and trachyte [23]. In that the rhyolite and trachyte outcrop in the nearby Euganean Hills [24], the only area of North Italy where these rocks can be found (rhyolite also in the Permian porphyry platform of the Trentino–South Tyrol region [25]), such subvolcanic products entail optimal lithic markers of provenance.

The main mineral components of the sand fraction transported by the Brenta and Bacchiglione are silica products (quartz—individual and polycrystalline grains—, chert and fossil shells), K-feldspars (orthoclase and microcline in Brenta and Adige sands, orthoclase and sanidine in Bacchiglione sands), plagioclase (chiefly Na-rich compositions) and carbonates (calcite and dolomite). The clay fraction consists mainly of montmorillonite, illite–montmorillonite, chlorite, illite and kaolinite [23].

## 1.2. The Earliest Foundry of Pre-Roman Padua and the Trade with the Adriatic Sea

The proto-urbanization in northern Italy took place at the beginning of the 1st millennium BCE, during the Early Iron Age. In the Veneto region, which already in the Final Bronze Age had experienced important processes of territorial rearrangement [26,27], land entities were based on large villages that were often located near a river. The proto-urban center of Padua grew up between the 9th and 8th centuries BCE along the bend and counter-bend of the Medoacus river [28]. The Medoacus river is now recognized as the ancient course of the Brenta river, later occupied by the current Bacchiglione [29,30].

One of the earliest known settlements of ancient Padua was located in the current Questura (police station), in the very center of the present-day city. This proto-urban site was strategically positioned along the course of the Medoacus river (Figure 1b) and was inhabited

during Roman, Medieval and Renaissance times. The area was partially excavated between 2000 and 2001, revealing structures related to craft activities (copper and pottery workshops) and residential areas datable from the late 9th to the 1st century BCE [31].

The excavation comprised a surface of 950 m² and two main areas were separated by a spoliation trench of Roman sewer oriented in an east-west direction (Figure 1c). The northern area was dedicated to art–craft activities and was mostly preserved, while the southern housing area was seriously damaged [32]. The materials excavated (ceramic sherds, metal artefacts, vegetal remains and animal bones) were offered for study by the Archaeological Superintendency of Veneto and are currently kept and fully accessible at the Archaeological Laboratories of the Department of Cultural Heritage of the University of Padua. The Early Iron Age pottery found at the Questura's site included fragments of typical pots in relation to Veneto culture and sherds with many calcareous inclusions, probably corresponding to an allochthonous ceramic class.

The literature data report many cases of ancient pottery tempered by carbonate inclusions, such as sparry calcite or fragments of carbonate rocks, speleothems and shells [20,33,34]. Sparry calcite, corresponding to rhombohedral in-shape crystals with cleavage planes, was found in the Veneto within veins from the Mesozoic and Tertiary limestones widely outcropping in the southern-alpine belt of the Veneto pre-Alps [35]. Speleothems are secondary deposits composed of $Ca_2CO_3$ formed in karstic caves and characterized by a saw-tooth pattern with alternation of white and thin dark-brown laminae [36,37].

Between the Final Bronze and the First Iron Ages, a specific ceramic class tempered with crushed fragments of speleothems spread across the hinterland of the Italian Friuli territory (north-eastern plains and Adriatic coast) and eastern Veneto region, with a progressive descent towards the Veronese area [38–41]. This class corresponded with a coarse ware production that was mainly used as containers for storage and exchange. In that pottery of this class has been found from the Final Bronze Age in Friuli Venezia Giulia and from later times in Veneto, it would seem to have been traded east-westwards [40,41]. Considering the strategic setting of Padua on the waterways connected with the mining districts of the Alps and the eastern Adriatic ports, such a ceramic class might have been used in a systematic trade of specific contents in the markets of the early city [41–43]. This interpretation is very important for provenance studies of pottery production, as karst environments are exceptional in Veneto (only in the Berici and Montello Hills, Figure 1a), while the Friuli region is a typically karstic area with clay-rich soil caves [44].

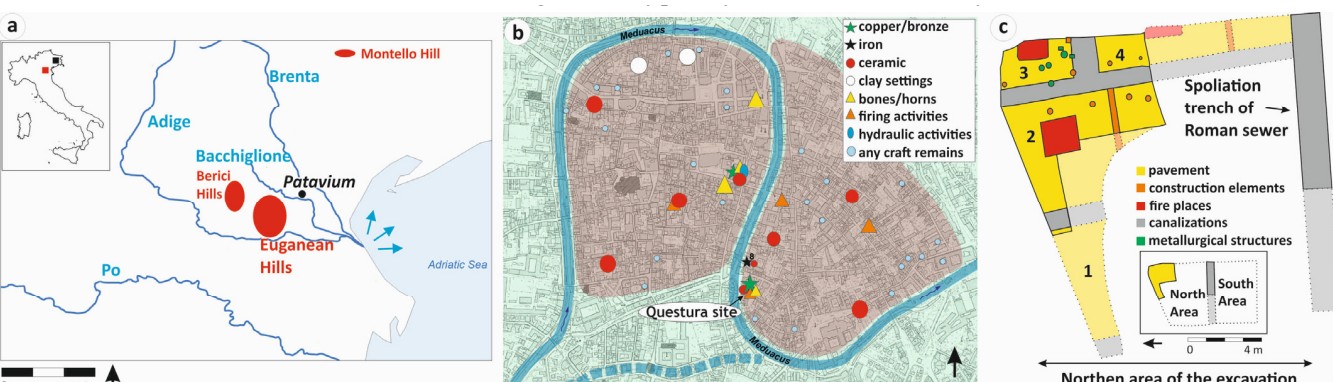

**Figure 1.** (**a**) Geographic location of Padua (Patavium in Roman times) and the Euganean, Berici and Montello Hills in the Veneto region (red square in the left insert) and the waterways of the main rivers flowing then nearby the city (from [45]). The Friuli Venezia Giulia region is also indicated (black square in the left insert); (**b**) Location of the Questura's site and other productive areas in the pre-Roman Padua during the 1st millennium BCE (from [31]); (**c**) sketch of the northern area of the Questura's site, highlighting the studied sectors (2, 3 and 4; the sherds analyzed in this paper were collected from sector 3), and schematic layout of the excavated area (from [46]).

## 2. Materials and Methods

### 2.1. The Pots from the Earliest Foundry of Pre-Roman Padua

The typological and chronological study of the pottery from the Early Iron Age found in the Questura's site was addressed by Ruzzante [32]. The author denominated each potsherd according to the stratigraphic units (US) where the ceramic sherd was found followed by the corresponding number of fragment. The potsherds collected during the archaeological field surveys performed in 2000–2001 subject of the analysis here presented came from the sector 3 of the productive northern area (Figure 1c).

Archaeometry analyses were carried out on a set of 40 fragments, corresponding to common wares of everyday pottery and to fine wares devoted to special occasions. The fine wares correspond to very delicate pieces (under 5 mm thickness).

The functional classes are cooking, table, storage and transport wares, both typical and extraneous to the Veneto repertoire. Any pot was wheel thrown [32] and the pottery is represented by the following morphology classes: 2 ollae, 2 cups, 2 turban-like bowls, 2 lids, 3 situliform vessels, 6 cups with raised handles, 5 biconical vessels, 6 bowls, 1 dolium (large container) and 6 pots (Figure 2). Besides the two ollae, two ollae handles were also analyzed. Six potsherds, mainly found within the US 1573, showed very abundant calcareous inclusions. In the present work, the denomination assigned by Ruzzante [32] to the potsherds corresponds with a progressive number—from 1 to 40—(Figure 2), that was used throughout the archaeometry study.

### 2.2. Multi-Analytical Approach

The study carried out comprised three methodological phases:

(i)   Visual examination and macroscopic description of the potsherds, both external and freshly cut surfaces.

(ii)  Petrographic analysis, mainly to describe micromass and porosity and to identify the minerals and rock fragments that constitute the inclusions. The 40 potsherds were studied in thin section by Polarizing Optical Microscopy (POM), using a Nikon Eclipse E660 microscope fitted with a CANON 650 digital camera and the Camera EOS digital microphotography system. The petrographic and textural features were described according to systematics proposed by Whitbread [47] and Quinn [48].

(iii) Spectroscopy analysis: on one side, mineralogical analyses of all the ceramic bodies were performed by means of X-ray Powder Diffraction (XRPD), in order to detect the mineral phases forming part of the ceramic bodies (pristine, firing and/or secondary phases). The external portion of each sherd was removed with a micro-drill to eliminate the material that could be differently manufactured and/or surface contamination. The ceramic bodies were reduced to powder in an agate mortar and then micronized (very fine powder, $\approx$10 μm) using a McCrone Micronising Mill [49]. X-ray Powder Diffraction data were obtained with a PANalytical X'Pert PRO diffractometer in Bragg–Brentano geometry equipped with a cobalt X-ray tube and a X'Celerator detector (40 kV voltage, 30 mA current, scanning interval 3–70°, equivalent step size 0.02° and equivalent counting time 1 s per step). The X'Pert HighScore Plus software was used to identify qualitatively mineral phases and XRPD data were statistically treated by cluster analysis according to the procedure proposed by Maritan et al. [50]. On the other, the chemical composition of the ceramic bodies of 29 sherds was determined by X-ray Fluorescence Spectrometry (XRF) on a WDS Panalytical Zetium sequential spectrometer. Beads were prepared from powder after calcination and mixed with $Li_2B_4O_7$, at a dilution ratio of 1:10. Quantitative chemical analyses of major and minor (wt% of $SiO_2$, $TiO_2$, $Al_2O_3$, $Fe_2O_3$, MnO, MgO, CaO, $Na_2O$, $K_2O$ and $P_2O_5$) and trace elements (ppm of S, Sc, V, Cr, Co, Ni, Cu, Zn, Ga, Rb, Sr, Y, Zr, Nb, Ba, La, Ce, Nd, Pb, Th and U) were carried out. Geological standards were used for calibration [51]. Loss on ignition (LOI) was determined heating the samples in a furnace at 860 °C for 20 min, and then at 980 °C for 2 h.

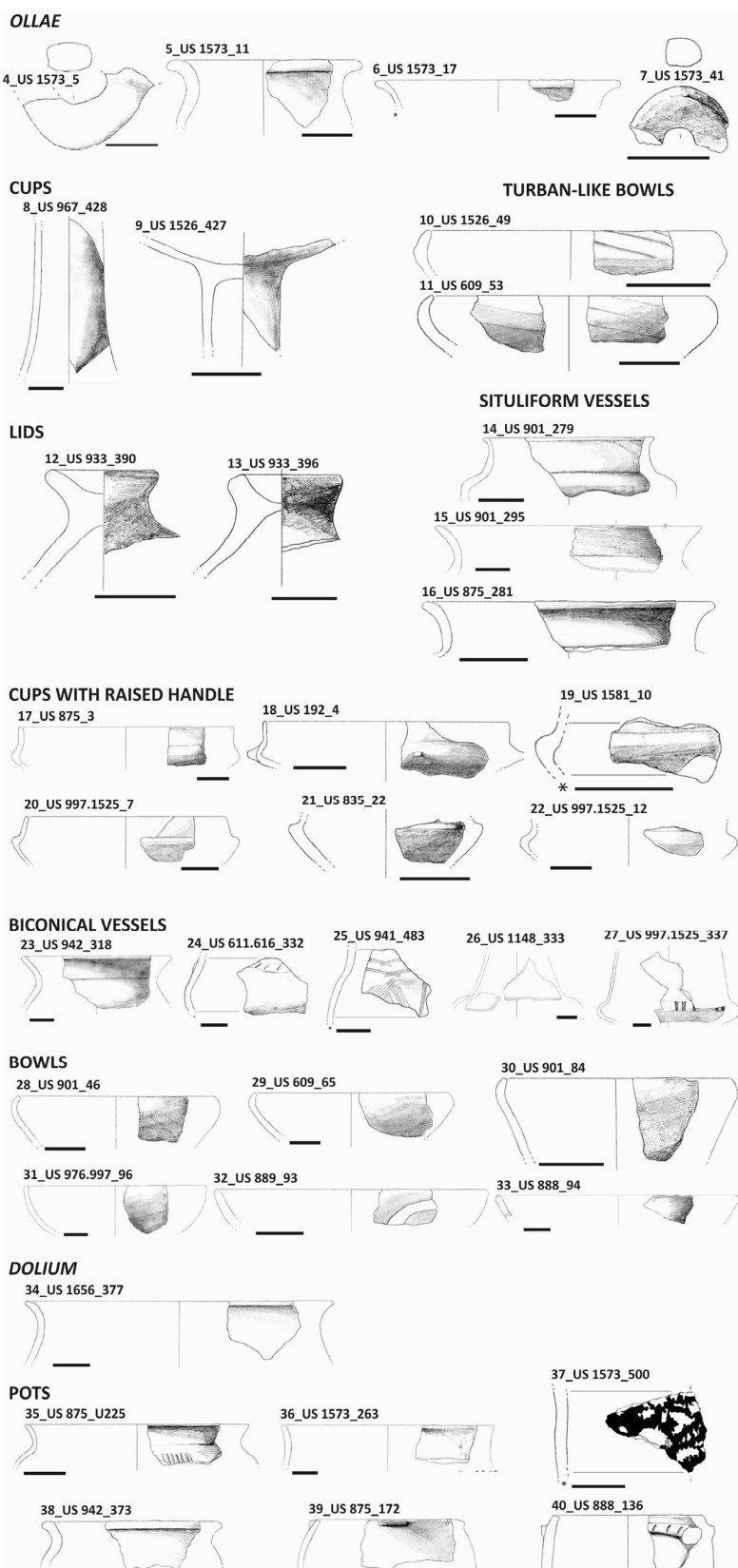

**Figure 2.** Morphological classes of the analyzed potsherds. US: stratigraphic units (US) from where the pots were found, according to Ruzzante [32]. No drawings of the morphologies of sherds named along the paper as n°1, 2 and 3 were available. Drawings by Ruzzante [32], scale bars added by authors (equivalent to 4 cm).

## 3. Results and Discussion

### 3.1. Macroscopic Description of External and Freshly Cut Surfaces

Many potsherds present very dark/dark-grey hue polished surfaces (sherds n°4, 7, 8, 11, 12, 14, 16–23, 28, 29 and 35) and some a light-brown hue smoothed surface (such as n°26, 30, 31, 37, 39 and 40). Other pots were just modeled, leaving slightly rough/rough surfaces (n°10). The highly smooth and very light-brown-colored material on the inner and outer faces of pot n°27 probably corresponds to a slip. Many pots display different color and/or finishing on both faces, maybe related to their functional class and/or long-time use. By closing the porosity, the polishing provided a water-proof resistance to the surfaces, according to the container use of the pots, that could be increased by the application of a specific waterproof material (black-colored material on surface of sherd n°37).

The surfaces of the pots exhibit scarce decorations, just braided reliefs (n°10), scratched ribbons (n°24, 25 and 35), strings (n°27), wavy line motifs (n°32) and cords (n°40). The sherds have a rather good conservation state, showing some discoloration and/or the partial loss of the polishing (in samples n°7 and 19, probably due to mechanical damage) and various incrustations, such as carbonaceous and highly oxidized deposits. It is worth mentioning the high porosity noted in the inner faces of samples n°5 and 25, with many calcareous inclusions, probably because of their detachment and/or chemical consumption provoked by the acid or aggressive contents that were contained within [32].

On the basis of the main common features observed on freshly cut cross sections, three main macro-groups were stated (Figure 3):

- Macro-group 1 (24 sherds) is composed by very dark-colored bodies with diverse grain-size inclusions: (i) heterogenous textured bodies with coarse-grained (2 mm—500 μm) and medium-grained (500—250 μm) inclusions; (ii) very uniform textured bodies (especially those shaping delicate pieces, ≈5 mm thick) with fine-grained (250—63 μm) inclusions and (iii) even textured bodies with many light, homogeneous in-size inclusions evenly spread. The inclusions are mainly coarse-grained (2 mm—500 μm) or medium-grained (500—250 μm), fine-grained inclusions (250—63 μm) only in sherd n°12.
- Macro-group 2 (6 sherds) corresponds to dark and brown-colored bodies with many coarse-grained and medium-grained carbonate inclusions, with angular/sub-angular shapes and rather uniformly distributed.
- Marco-group 3 (10 sherds) comprises a miscellany of pots with very dark or brown-colored bodies (towards an orange hue in sherds n°23 and 34). The inclusions are different in type, abundance and size.

The very dark color of many ceramic bodies may point out the presence of organic matter in the starting raw clays, which induced a reducing environment of firing then preserved while cooling [9,52]. The plentiful inclusions evenly distributed within some sherds of macro-group 1 and sherds of macro-group 2 were probably added (tempered pots). A low porosity is noted, except in n°13 and 40—somehow fissured due to decay processes—and in n°34 and 38, with elongated pores formed during the manufacturing.

Many dark-colored bodies of macro-group 1 show an external border with the same light-brown hue as the one observed in the external surfaces. As the dark color of ceramic bodies can be turned into a red color by oxidation at the end of firing and/or during a long cooling [52], this light-brown color could have been intentionally pursued by oxidation, maybe for aesthetic purposes. The very dark external border observed in some brown-colored bodies of the miscellaneous group (sherds n°14, 21 and 23), may suggest the adoption of reduction conditions at the end of the firing [1]. Specific features were observed in some of the miscellaneous sherds, such as a very fine thickness (2–5 mm thick in pots n°14, 21, 27 and 35), a granular texture (n°17), a blurred appearance (n°14, 21 and 35) or a light-grey hue band (2 mm thick) parallel to the inner and outer faces (n°27). Besides, sherd n°14 shows a light-brown-hued external border (1–2 mm thick) that may point out the use of a specific practice to increase the hardness of such a delicate piece.

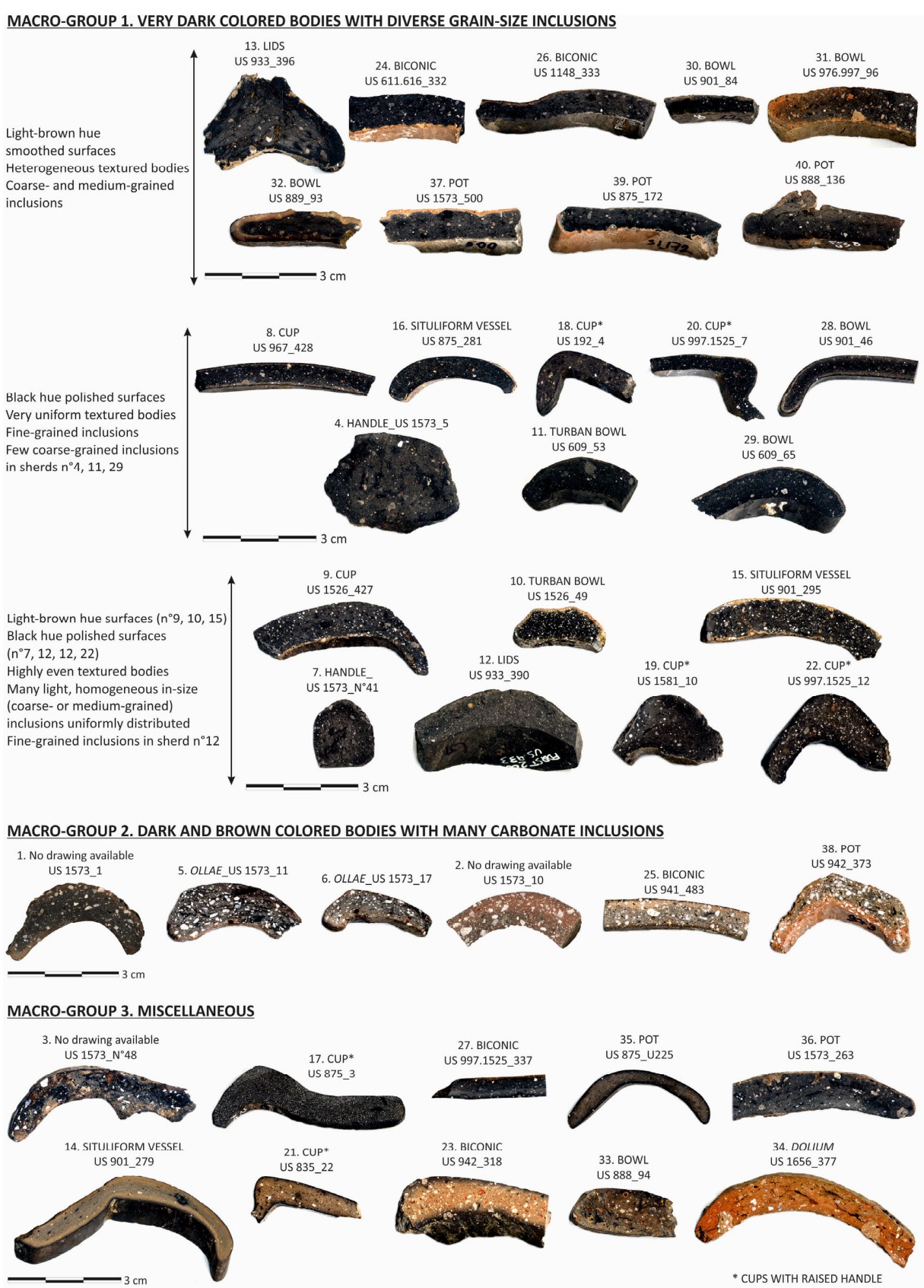

**Figure 3.** Photographs of the ceramic body of the studied potsherds as taken from the freshly cut surfaces. Three groups were stated on the basis of the main macro-textural features. Some characteristics of the external surfaces of the pots can be also observed.

*3.2. Petrographic Features*

Two main petro-fabrics and miscellaneous petro-fabrics were defined under POM, in concordance with the macroscopic observations.

3.2.1. Petro-Fabric 1: Potsherds with Silicate Inclusions (24 Sherds)

This group comprises ceramic pastes characterized by a dark-brown, uniform and optically inactive groundmass in which are embedded mainly silica-rich inclusions (conspicuous quartz and K-feldspars). Elongated pores are noted, pointing out that low firing temperatures were reached. The only traces of high firing temperatures are scarce transformations at grain boundaries and reaction rims formed around a few inclusions, just in the over-fired sherds (n°7 and 19). According to the diverse quantity and size of the inclusions, three sub-fabrics were distinguished.

Petro-Fabric 1a: Uneven Textured Groundmass with Coarse- and Medium-Grained Inclusions (Sherds n°13, 24, 26, 30–32, 37, 39 and 40)

This petro-fabric is characterized by rather porous pastes with 30–40% of grains very heterogeneous in size. These grains correspond to fragments of polycrystalline quartz, metamorphic, volcanic (trachyte groundmass) and argillaceous rock fragments (ARF), carbonate inclusions (mainly sub-rounded grains of micritic limestone), as well as Fe-rich clay pellets and grog (Figure 4a–d). The light-brown hue of the external border (1 in Figure 4c) is due to the oxidizing conditions adopted at the end of firing, as stated. The porosity mainly corresponds to cracks, to shrinking rims formed around and inside the coarse-size grains (Figure 4a) due to their contraction during the cooling and to channel voids with dark boundaries (Figure 4c) resulting from the decomposition of organic inclusions.

Petro-Fabric 1b: Even Textured Groundmass with Fine-Grained Inclusions (Sherds n°8, 16, 18, 20 and 28)

Very low porous pastes with 5% fine-grained inclusions with unimodal grain-size distribution (Figure 4e–h), mainly composed of quartz associated with feldspars and occasional pyroxene crystals. A few coarse-size fragments of polycrystalline quartz and trachyte with oriented texture also occur (Figure 4g). Sherds n°4, 11 and 19 belong to this sub-fabric, also containing some coarse-size inclusions.

Petro-Fabric 1c: Highly Even Textured Groundmass with Silicate Tempering (Sherds n°7, 9, 10, 12, 15, 19 and 22)

Potsherds with abundant inclusions (40%) and bimodal grain-sized distribution. The coarse-grained inclusions mostly correspond to sub-rounded polycrystalline quartz grains and metamorphic (chiefly quartzite and mica–schist) and volcanic (trachyte with oriented texture and rhyolite) rock fragments (sherds n°9, 10 and 15, Figure 4i,j). Some pyroxene crystals, fragments of grog and shells and rare micritic limestone grains also occur. The medium-grained inclusions (n°7, 19 and 22) are angular/sub-angular crystals of quartz and felspars and sub-rounded micritic limestone grains (samples n°7, 12, 19 and 22, Figure 4k,l). The coarse- and medium-size fractions are well sorted and both are evenly distributed.

3.2.2. Petro-Fabric 2: Potsherds with Calcite Temper (6 Sherds: n°1, 2, 5, 6, 25 and 38)

This petro-fabric is characterized by abundant/very abundant (20–40%) carbonate inclusions, especially in sherd n°38 (50%), showing a polymodal grain-sized distribution. These inclusions are chiefly composed of coarse-size crystals of sparry calcite and of sub-angular fragments of speleothems that are even larger than 2 mm in size (Figure 5). Some sparite and micrite limestone fragments and fine-grained quartz crystals (samples n°2 and 25) also occur. The dark-brown hue groundmass is optically inactive (only with slight optical activity in n°25) and likewise shows elongated pores (Figure 5a,e).

**MACRO-GROUP 1 = Petro-fabric 1: potsherds with silicate inclusions**
**Inclusions: mainly crystals of quartz and K-feldspars and rock fragments of trachyte, rhyolite, quartzite and mica-schist**
<u>Petro-fabric 1a</u>: uneven textured groundmass with coarse- and medium-grained inclusions

<u>Petro-fabric 1b</u>: very uniform groundmass with fine-grained inclusions

<u>Petro-fabric 1c</u>: highly even textured groundmass with silicate tempering
Coarse-size temper: sub-rounded grains of polycrystalline quartz and fragments of quartzite, mica-schist, trachyte and rhyolite
Medium-size temper: angular/sub-angular quartz and feldspars crystals and sub-rounded micritic limestone grains

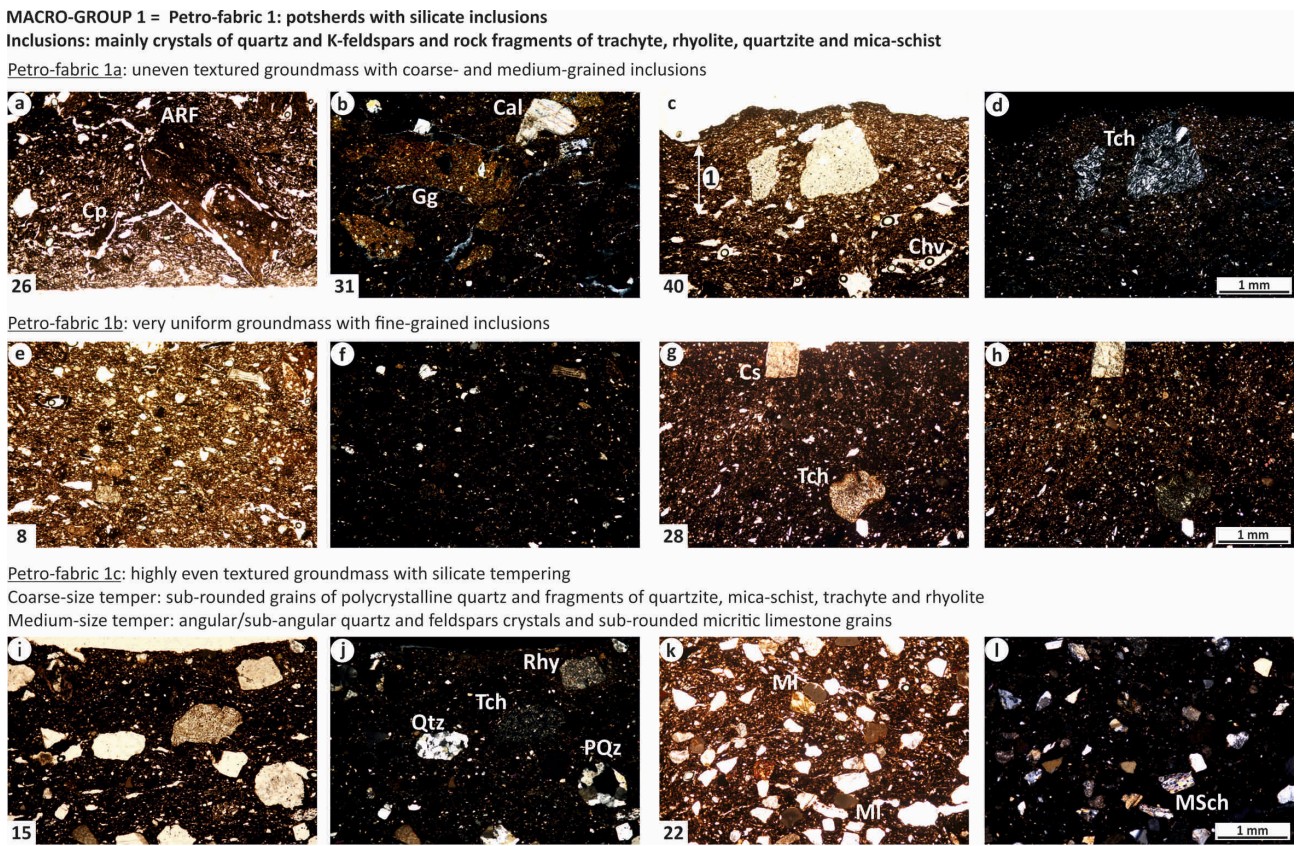

**Figure 4.** Photomicrographs in plane-polarized light (**a**,**c**,**e**,**g**,**i**,**k**) or crossed-polarized light (**b**,**d**,**f**,**h**,**j**,**l**) of ceramic bodies belonging to petro-fabric 1. Light-brown hue external border formed under oxidizing conditions at the end of firing (1 in (**c**)). Abbreviations: ARF: argillaceous rock fragment, Cp: clay pellet, Cal: calcite, Gg: grog, Chv: channel void, Tch: trachyte, Cs: sparry calcite, Qtz: quartzite, PQz: polycrystalline quartz, Rhy: rhyolite, Ml: micritic limestone, MSch: mica-schist. The scale bar is the same for all the images.

**MACRO-GROUP 2 = Petro-fabric 2: calcite-tempered potsherds**
**Temper: coarse-grained crystals of sparry calcite and of sub-angular fragments of speleothems (even > 2 mm)**

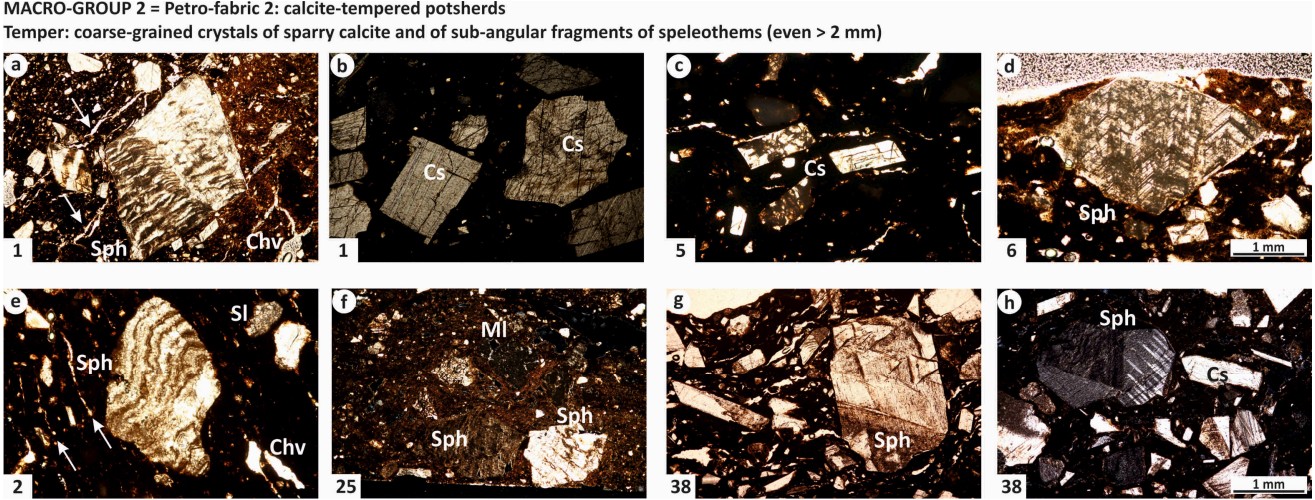

**Figure 5.** Photomicrographs in plane-polarized light (**a**,**c**,**d**,**e**,**g**) or crossed-polarized light (**b**,**f**,**h**) of the ceramic bodies belonging to petro-fabric 2. Elongated pores marked with arrows (**a**,**e**). Abbreviations: Sph: speleothems, Chv: channel void, Cs: sparry calcite, Sl: sparite limestone, Ml: micritic limestone. The scale bar is the same for all the images.

### 3.2.3. Miscellaneous Potsherds (10 Sherds: n°3, 14, 17, 21, 23, 27, 33–36)

Sherd n°3 displays 10% of inclusions, corresponding chiefly with medium- and coarse-grained crystals of sparry calcite (Figure 6a). Paste n°17 contains very abundant (60%) fine-grained inclusions, very well sorted and with unimodal grain-sized distribution (Figure 6b,c). They are composed of sub-angular quartz crystals and polycrystalline quartz grains and of sub-rounded mica–schist, trachyte, rhyolite and micritic limestone fragments. The micromass of n°35 (with a blurred appearance on the freshly cut surface) is optically active and the inclusions are abundant (40%) and well sorted, mainly corresponding with very fine sorted sand. The groundmass of n°27 is also optically active and the fine-grained inclusions are composed of dominant quartz and phyllosilicate. A few medium-size inclusions (5–10%) also occur, corresponding to micritic limestone grains with irregular shapes associated with clay pellets, Fe-rich grains and grog. As observed in the hand specimen, a slightly lighter area is observed (2 in Figure 6d), but no other specific differences were detected.

**MACRO-GROUP 3. Miscellaneous petro-fabrics**

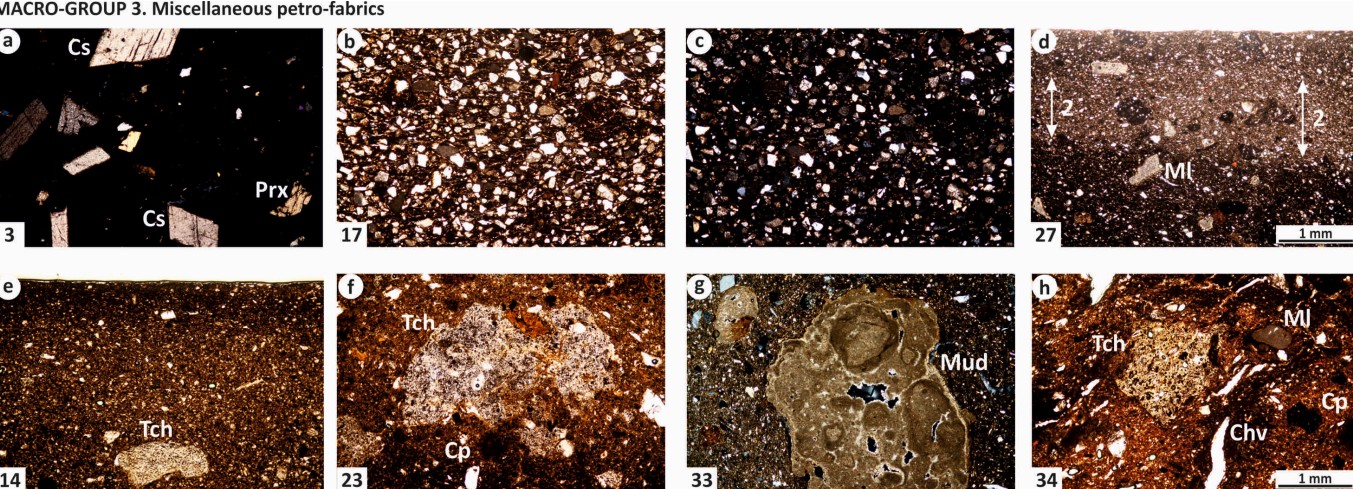

**Figure 6.** Photomicrographs in plane-polarized light (**b,d,e,f,h**) and crossed-polarized light (**a,c,g**) of the ceramic bodies belonging to the miscellaneous group. A slightly lighter area is observed within sample n°27 (2 in (**d**)). Abbreviations: Cs: sparry calcite, Prx: pyroxene, Ml: micritic limestone, Tch: trachyte, Cp: clay pellet, Mud: mudstone, Chv: channel void. The scale bar is the same for all the images.

The brown-colored bodies n°14 and 21, also with a blurred appearance on freshly cut surfaces, are characterized by an optically active groundmass. They display many inclusions (50–60%), very fine sorted sand composed predominantly by quartz, feldspars and phyllosilicates. Some K-feldspar crystals and rare millimetric fragments of trachyte with oriented texture also occur in sample n°14 (Figure 6e). A few sparry calcite crystals and fragments of trachyte, micritic limestone grains and Fe-rich clay pellets also occur in n°21.

The samples n°23, 33 and 34 show abundant inclusions (30%) with polymodal grain-size distribution and many millimetric grains. Body n°23 has a groundmass slightly optical active and displays profuse iron oxides and Fe-rich clay pellets. A few millimetric fragments of trachyte also occur, some of them partially reacted with the evolving Fe-rich groundmass (Figure 6f). The micromass of sherd n°33 is also to some extent optically active and is characterized by millimetric and sub-rounded mudstone grains (Figure 6g). Pyroxene crystals and fragments of grog, trachyte and mica–schist also occur. Sample n°34 has a highly porous and heterogeneous in color groundmass with dark- and light-red (slightly optically active) areas. The coarse-size inclusions are constituted by polycrystalline quartz, pyroxene and plagioclase crystals, Fe-rich clay pellets and iron oxides, as well as by fragments of trachyte and argillaceous rocks and carbonate grains (micrite limestone and speleothems fragments). Rather abundant channel voids with dark boundaries were formed (Figure 6h).

As fragments of rhyolite, quartzite and mica–schist are rather more abundant in the Adige plain deposits [23,53], their conspicuous presence within the pots belonging to petro-fabric 1 may suggest the use of raw clays from the sediments transported by the Adige river, as observed in other pottery productions nearby Padua [22]. More specifically, this river flows through the southern area of the Euganean Hills, where trachyte with oriented texture is typically outcropped [45].

Any specific treatment of the base clays seems that was carried out to produce the pots belonging to petro-fabric 1a, probably because such pots corresponded with dairy common ware. The shrinking cracks around the coarse-size grains triggered in turn the fissuring of the ceramic bodies (Figure 4a). The production of fine wares accomplished two main technological choices: (i) the purification of the starting raw clays; hence, fine/very fine-grained clay pastes were achieved (pots belonging to petro-fabrics 1b and 1c), In fact, clay settling pools from the Early Iron Age have been found near the Questura's site [32], and (ii) the tempering of the clay pastes with diverse grain-size inclusions attained probably from the sieving of the starting clayey materials, adding natural sediments (with sub-rounded shapes, being therefore sand) or grounded (sub-angular in-shape) temper.

The types of inclusions, especially the trachyte rock fragments, observed in many sherds of the miscellaneous group (n°14, 17, 21, 23, 34 and 35) point out the selection of similar clayey materials to those used to produce pots of macro-group 1. The millimetric coarse-size mudstone grains detected in body n°33 and the coarse-size micritic limestone grains with irregular shapes of sample n°27 suggest that clayey deposits from another area were used.

The diverse type and size and, overall, the different quantity of carbonate temper shown by the calcite-tempered pots may point out that they corresponded to different productions. Some of the miscellaneous sherds were produced with the same technological choices adopted for the manufacturing of pottery from petro-fabrics 1b and 1c. Therefore, the purification of the raw clays was also carried out to produce sherds n°14, 21 and 35, which were in turn fired at low temperatures (below the breakdown of phyllosilicate, as denotes the optically active groundmass). Likewise, the clayey paste of sherd n°17 was tempered with grains both obtained from the sieving of the base clayey materials and grounded. In this case, much more quantity of temper was added.

*3.3. Chemical Composition*

The XRF results of the 29 potsherds analyzed are presented in Table 1. The bodies of the potsherds with silicate inclusions (petro-fabric 1) are chemically very similar. They are very rich in silica ($\overline{X}$ = 63.2 wt%) and alumina ($\overline{X}$ = 18.1 wt%), moderately rich in iron oxide ($\overline{X}$ = 6.36 wt%) and alkali ($\overline{X}Na_2O$ = 1.67 wt%, $\overline{X}K_2O$ = 2.33 wt%) and have low-medium contents in alkaline–earth metals (CaO + MgO from 3 to 8%). These data point out that the base clays were silica-rich (high $SiO_2$ values) with an important clay mineral content (high $Al_2O_3$ and $Fe_2O_3$ and important $K_2O$ concentrations) and rather rich in K-feldspars (important $K_2O$ amounts). The average content of $TiO_2$ (1.44 wt%) is related to the Ti-bearing accessory minerals commonly present in the clayey materials of the Veneto region [23]. The detection of quantities of $P_2O_5$ about 1 wt% may point out some contamination processes due to the burial of the sherds, since the content of $P_2O_5$ in the clay deposits from Veneto normally varies between 0.05 and 0.50 wt% [54]. The uniform chemical composition of the bodies of the petro-fabric 1 is reflected by the low std values, somewhat higher on petro-fabric 1c and probably related with the silica-rich tempering.

**Table 1.** Major and minor oxides (in wt%) of the ceramic pastes determined by XRF.

**Petro-Fabric 1. Potsherds with Silicate Inclusions**

| | n° | SiO$_2$ | TiO$_2$ | Al$_2$O$_3$ | Fe$_2$O$_3$ | MnO | MgO | CaO | Na$_2$O | K$_2$O | P$_2$O$_5$ | Tot | LOI * |
|---|---|---|---|---|---|---|---|---|---|---|---|---|---|
| | 13 | 64.6 | 1.42 | 18.3 | 6.32 | 0.08 | 1.78 | 2.24 | 1.75 | 2.30 | 0.67 | 99.5 | 7.64 |
| Petro-fabric 1a | 24 | 61.0 | 1.64 | 18.0 | 7.18 | 0.08 | 2.23 | 3.83 | 1.43 | 2.36 | 1.30 | 99.0 | 8.50 |
| | 30 | 62.7 | 1.55 | 17.7 | 6.44 | 0.05 | 1.58 | 3.73 | 1.57 | 2.21 | 1.84 | 99.4 | 9.95 |
| | 39 | 66.5 | 1.37 | 17.4 | 5.78 | 0.05 | 1.19 | 2.19 | 1.87 | 2.35 | 0.79 | 99.5 | 4.78 |
| | 18 | 63.0 | 1.31 | 20.5 | 5.38 | 0.07 | 1.21 | 2.91 | 1.89 | 2.47 | 0.65 | 99.4 | 11.0 |
| Petro-fabric 1b | 20 | 64.4 | 1.42 | 17.9 | 6.54 | 0.12 | 1.75 | 3.36 | 1.58 | 2.49 | 0.74 | 100.3 | 4.48 |
| | 28 | 64.7 | 1.51 | 18.2 | 5.81 | 0.05 | 1.38 | 2.89 | 1.54 | 2.01 | 1.39 | 99.5 | 9.37 |
| | 11 | 62.8 | 1.56 | 18.2 | 7.08 | 0.08 | 1.91 | 3.02 | 1.53 | 2.26 | 0.64 | 99.1 | 5.98 |
| | 9 | 61.8 | 1.02 | 19.4 | 5.13 | 0.05 | 1.16 | 4.46 | 2.31 | 2.78 | 1.19 | 99.3 | 9.78 |
| | 10 | 67.5 | 0.89 | 16.6 | 5.45 | 0.07 | 1.77 | 2.70 | 1.59 | 2.61 | 0.47 | 99.7 | 2.96 |
| Petro-fabric 1c | 15 | 63.5 | 1.31 | 17.7 | 6.04 | 0.05 | 1.67 | 3.34 | 1.45 | 2.19 | 1.30 | 98.5 | 10.0 |
| Si-rich | 7 | 59.2 | 1.98 | 19.2 | 8.42 | 0.12 | 2.36 | 3.62 | 1.80 | 2.24 | 0.43 | 99.4 | 1.61 |
| tempering | 12 | 62.3 | 1.39 | 18.2 | 5.96 | 0.05 | 1.97 | 4.43 | 1.55 | 2.23 | 1.39 | 99.4 | 10.8 |
| | 19 | 63.6 | 1.41 | 17.8 | 6.79 | 0.12 | 1.75 | 3.05 | 1.54 | 2.43 | 0.44 | 99.0 | 2.95 |
| | 22 | 60.8 | 1.82 | 17.0 | 7.01 | 0.06 | 2.76 | 4.92 | 1.70 | 2.03 | 1.35 | 99.4 | 11.3 |
| | X̄ | 63.23 | 1.44 | 18.14 | 6.36 | 0.07 | 1.76 | 3.38 | 1.67 | 2.33 | 0.97 | 99.36 | 7.41 |
| | Std | 2.16 | 0.27 | 0.97 | 0.86 | 0.03 | 0.45 | 0.80 | 0.23 | 0.20 | 0.44 | 0.39 | 3.32 |

**Petro-Fabric 2. Calcite-Tempered Potsherds**

| | n° | SiO$_2$ | TiO$_2$ | Al$_2$O$_3$ | Fe$_2$O$_3$ | MnO | MgO | CaO | Na$_2$O | K$_2$O | P$_2$O$_5$ | Tot | LOI |
|---|---|---|---|---|---|---|---|---|---|---|---|---|---|
| | 1 | 55.1 | 0.79 | 14.2 | 4.85 | 0.08 | 1.68 | 19.4 | 0.66 | 1.59 | 0.72 | 99.1 | 19.3 |
| | 5 | 50.0 | 0.68 | 14.2 | 4.23 | 0.03 | 1.14 | 26.2 | 0.16 | 1.97 | 0.56 | 99.1 | 18.6 |
| Dark-colored | 6 | 52.0 | 0.71 | 15.8 | 5.30 | 0.04 | 1.03 | 21.1 | 0.30 | 2.27 | 0.80 | 99.3 | 16.0 |
| bodies | X̄ | 52.4 | 0.73 | 14.7 | 4.79 | 0.05 | 1.28 | 22.2 | 0.37 | 1.94 | 0.69 | 99.2 | 18.0 |
| | Std | 2.57 | 0.06 | 0.92 | 0.54 | 0.03 | 0.35 | 3.54 | 0.26 | 0.34 | 0.12 | 0.12 | 1.74 |
| | 2 | 44.3 | 0.86 | 15.1 | 4.01 | 0.05 | 1.66 | 30.6 | 0.23 | 1.56 | 1.20 | 99.6 | 24.2 |
| | 25 | 49.8 | 0.75 | 14.7 | 4.15 | 0.05 | 0.94 | 26.2 | 0.16 | 1.89 | 0.80 | 99.4 | 19.5 |
| Brown-colored | 38 | 40.3 | 0.67 | 15.3 | 4.05 | 0.03 | 1.41 | 34.3 | 0.28 | 2.31 | 0.45 | 99.0 | 22.6 |
| bodies | X̄ | 44.8 | 0.76 | 15.0 | 4.07 | 0.04 | 1.34 | 30.4 | 0.22 | 1.92 | 0.82 | 99.3 | 22.1 |
| | Std | 4.77 | 0.10 | 0.31 | 0.07 | 0.01 | 0.37 | 4.06 | 0.06 | 0.38 | 0.38 | 0.31 | 2.39 |

**Miscellaneous Petro-Fabrics**

| | n° | SiO$_2$ | TiO$_2$ | Al$_2$O$_3$ | Fe$_2$O$_3$ | MnO | MgO | CaO | Na$_2$O | K$_2$O | P$_2$O$_5$ | Tot | LOI |
|---|---|---|---|---|---|---|---|---|---|---|---|---|---|
| | 3 | 58.6 | 1.55 | 17.3 | 6.76 | 0.05 | 2.18 | 8.56 | 1.66 | 2.34 | 0.64 | 99.6 | 9.71 |
| | 17 | 58.2 | 0.92 | 15.6 | 4.81 | 0.07 | 4.70 | 11.3 | 0.84 | 2.05 | 1.11 | 99.6 | 16.2 |
| | 27 | 61.8 | 0.93 | 21.2 | 6.48 | 0.09 | 1.93 | 2.31 | 1.12 | 3.64 | 0.39 | 99.8 | 2.63 |
| | 35 | 57.9 | 0.79 | 21.3 | 5.86 | 0.08 | 1.96 | 6.42 | 0.87 | 3.46 | 0.67 | 99.2 | 12.8 |
| | 36 | 64.9 | 1.15 | 18.1 | 5.38 | 0.04 | 1.45 | 3.68 | 1.52 | 2.56 | 0.54 | 99.3 | 7.56 |
| | 14 | 63.8 | 0.97 | 18.0 | 5.76 | 0.05 | 1.40 | 2.31 | 2.14 | 3.00 | 1.47 | 98.9 | 7.69 |
| | 21 | 65.4 | 0.90 | 17.8 | 5.15 | 0.04 | 1.49 | 2.05 | 2.36 | 3.20 | 0.98 | 99.4 | 7.03 |
| | 23 | 62.5 | 1.25 | 19.4 | 5.25 | 0.05 | 0.09 | 3.11 | 2.39 | 3.30 | 2.29 | 99.7 | 9.64 |
| | X̄ | 61.64 | 1.06 | 18.59 | 5.68 | 0.06 | 1.90 | 4.97 | 1.61 | 2.94 | 1.01 | 99.44 | 9.16 |
| | Std | 3.05 | 0.25 | 1.95 | 0.67 | 0.02 | 1.30 | 3.44 | 0.64 | 0.57 | 0.62 | 0.30 | 4.06 |

* Loss on ignition.

The calcite-tempered sherds n°1, 5 and 6 (with dark-colored bodies) show analogous chemical composition: rather high silica ($\overline{X} \approx 52$ wt%) and alumina content ($\overline{X} \approx 15$ wt%), important iron oxide ($\overline{X} \approx 5$ wt%) and very high CaO contents ($\overline{X} \approx 22$ wt%). Those with brown-colored pastes (sherds n°2, 25 and 28) were found to have even higher CaO contents ($\overline{X} \approx 30$ wt%, especially high in sample n°38 = 34.4 wt%) but are lower in silica and iron oxide ($\overline{X} \approx 45$ wt% and $\approx 4$ wt%, respectively). These data point out that the raw clays used to produce the calcite-tempered pots had lower silica and clay mineral contents than those of the clayey materials used for manufacturing the pots of petro-fabric 1. Although CaO contents over 6 wt% point out the use of calcareous-rich clays [55], such high concentrations can be attributed to the significant quantity of calcite temper added to the clay pastes. The very high LOI values (>16 wt%) are mostly due to the decarbonization of the abundant calcite inclusions present in the sherds of petro-fabric 2, as the $CO_2$ is lost after calcination.

Regarding the miscellaneous sherds, the high CaO content ($\approx 8.6$ wt%) detected in sample n°3 is chiefly because of the significant presence of calcite inclusions (sparry calcite). The high quantities of CaO (11.3 wt%) and MgO (4.7 wt%) in sample n°17 (tempered pot) suggest that the temper corresponding to micritic limestone grains should have a calcitic and/or dolomitic composition. Samples n°14 and 21 show a very similar chemical composition to the pots of petro-fabric 1; just the concentrations of iron oxide are to some extent lower and of alkaline–earth oxides slightly higher. Sample n°23 has also rather analogous oxide content to sherds of petro-fabric 1, though the alkali oxides are higher and almost no MgO concentration was detected.

From the plotting of the chemical data on the triangular diagrams shown in Figure 7, the grouping of sherds with silicate inclusions and with calcite tempering is clearly defined. Bodies with silicate inclusions belonging to petro-fabric 1 and to the miscellaneous group were diversely plotted, many of them rather jointly (continuous line). The dark-colored samples n°3, 17, 27 and/or 35 (miscellaneous) were normally plotted very scattered. On the basis of the $SiO_2$-$Al_2O_3$-CaO contents (Figure 7a), bodies of petro-fabric 1 and sherds n°14, 21 and 23 were plotted jointly gathered, which silica and alumina content just within those of the clay minerals. When CaO, $Fe_2O_3$ and MgO concentrations are considered (Figure 7b), the data present a higher dispersion. If the main oxides forming part of the silicate inclusions and clay minerals are considered and the lime content is removed (Figure 7c), samples n°1 (calcite-tempered pot) and 17 (silicate- and carbonate-tempered pot) present almost the same plotting outside of the two main groups (nearer from the bodies with silicate inclusions, dotted line). As Na-plagioclase and K-feldspars are abundant and Ti-bearing accessory minerals are commonly present in the sediments transported by the rivers flowing near Padua [23], the contents of $K_2O$, $Na_2O$ and $TiO_2$ have also been plotted (Figure 7d). The high scattering of some bodies with silica inclusions was likewise achieved and sherds n°1 and 17 were plotted again rather jointly.

Regarding the trace elements, sherds belonging to petro-fabric 1 display likewise rather homogeneous contents. However, very different quantities of S, Zn and Cu were detected. The dissimilar contents of S and Zn could be due to the contamination of sherds because of alteration processes that might take place in the burial environment where they were found. The divergent contents of copper are probably related with the metal production accomplished in the workshops that once were in the site. Many sherds belonging to the miscellaneous group show significant chemical affinity with pots of petro-fabric 1 respecting to the content of trace elements, mainly regarding V, Ga, La and Th quantities and, to a lesser extent, concerning Cr, Co, Ni, Zn and Rb contents.

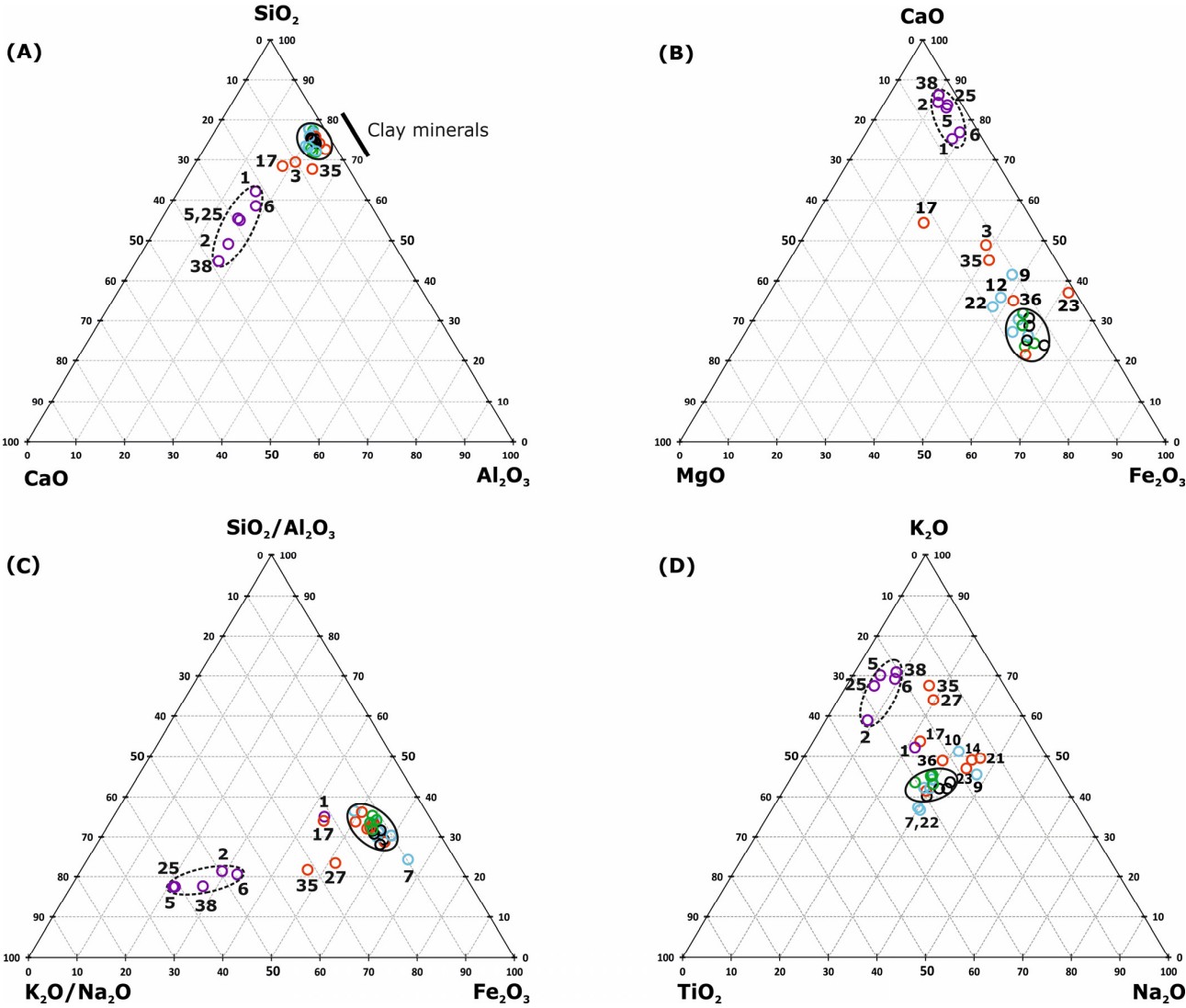

**Figure 7.** (**A–D**) Ternary diagrams of some selected oxides or oxide ratios. Calcite-tempered bodies show almost the same degree of clustering (dotted line) while bodies with silicate inclusions were diversely plotted in the various diagrams, many of them rather jointly (continuous line, included within potsherds belonging to petro-fabric 1 and miscellaneous). Petro-fabric 1 (potsherds with silicate inclusions): 1a (black circles), 1b (green circles) and 1c (Si-rich tempering, blue circles); Petro-fabric 2 (purple circles); Miscellaneous (red circles). The dark-colored bodies n°3, 17, 27 and/or 35 are normally plotted very scattered.

In the score and loading plots of principal component analysis (PCA), bodies with silicate inclusions, belonging both to the petro-fabric 1 and the miscellaneous group, clearly differ from calcite-tempered sherds. Therefore, the last show a higher content of Ca and lower contents of Al, Si, Na and Fe (Figure 8). Some sherds of petro-fabric 1a, almost all belonging to petro-fabric 1c (except sample n°10) and sherd n°3 (miscellaneous) plot at positive values of PC1 and negative of PC2, and they are highly rich both in Si, Na and Fe and Sr, Ti, Zr, Nd and Nb. The bodies n°14, 20, 21, 36 and 39 plot rather jointly at positive values of PC1 and PC2, with high values of Al and Ba as well as of Th, Ga, U and La. According to the loading plots, sample n°10 is rich in Cu, and samples n°27 and 35 are highly rich in K, Rb and Pb.

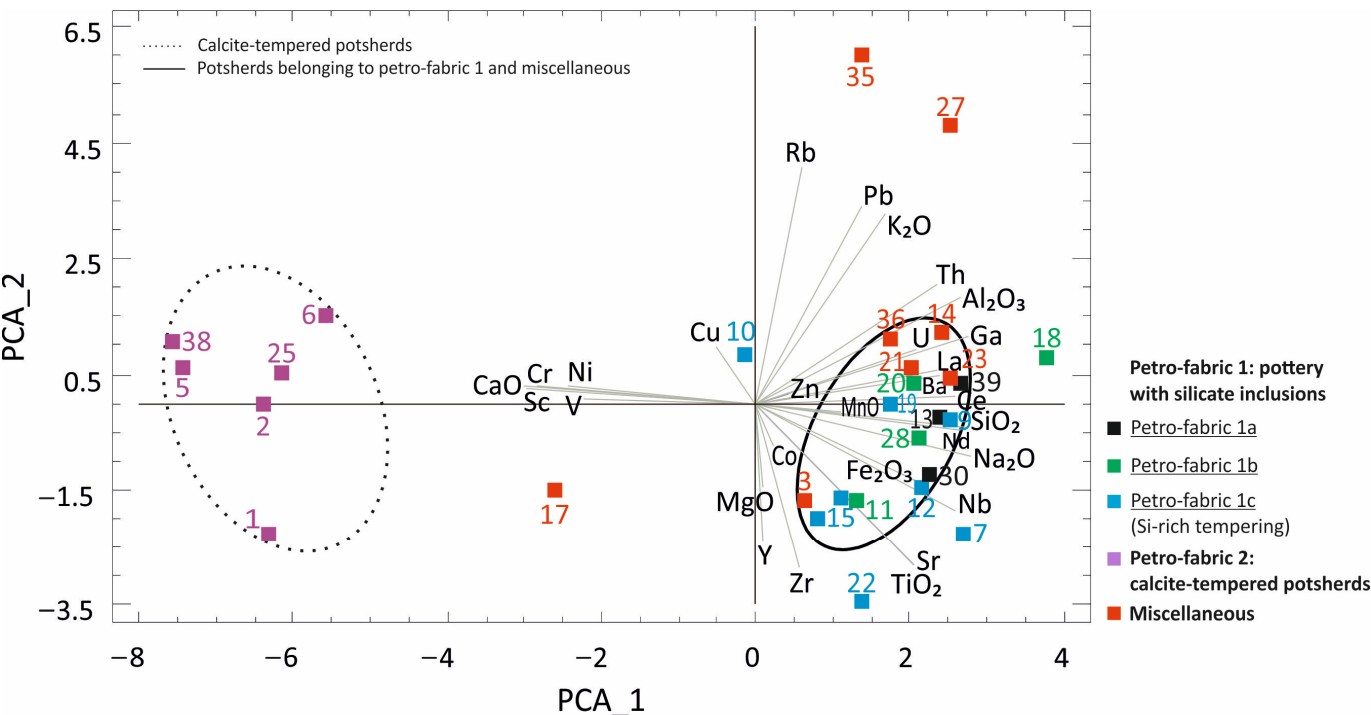

**Figure 8.** Score and loading plots of PC1 and PC2, representing 45% and 13% of total variance, respectively, obtained by PCA of chemical data of 29 potsherds. The calcite-tempered bodies (dotted line) and many bodies with silicate inclusions (continuous line) display clear differences regarding the contents of Ca, Al, Si, Na and Fe. The dark-colored bodies n°27 and 35 are rich in Rb, Pb and K.

The very different chemical compositions of the pots belonging to petro-fabric 1 with respect to the calcite-tempered pots (petro-fabric 2) points out the use of very different raw clays. Differences observed among samples of petro-fabric 2 are related to the content of calcite but also to the use of different base clays (even if not so dissimilar), since for instance potsherd n°1 clearly differ from n°6 in terms of trace elements. The miscellaneous pots n°14 and 21 were produced with raw clays compositionally very similar, in turn rather similar to those used for pottery belonging to petro-fabric 1. Pots n°17, 27 and 35 were produced with clays from different areas, most probably also in the Veneto region, that were collected from similar deposits but differing in terms especially of carbonate content, so that they are closer to the set of samples of petro-fabric 2, obtained with the addition of calcite to the base clay (with very high CaO content). Pot n°10 may correspond, likewise, to a specific regional production; in fact, it is a unique sherd showing a highly rough surface intentionally made, probably related with metal production activities, hence the copper enrichment detected.

### 3.4. Mineralogical Composition

The mineralogical assemblages detected by XRPD in the ceramic bodies are shown in Table 2. Sherds of petro-fabric 1 have a very similar mineralogical composition, corresponding to predominant quartz and subordinated albite and/or illite and to diverse concentrations of K-feldspars (mostly sanidine). Besides, chlorite was occasionally present (sherds n°12, 15, 18 and 22) and calcite and/or dolomite were detected in low quantities in samples n°18, 20 and 24 (petro-fabric 1b) and in n°10, 12, 19 and 22 (petro-fabric 1c). Pretty homogeneous mineral association was found within the majority of the carbonate-tempered bodies (petro-fabric 2), corresponding to predominant calcite (especially abundant in sample n°38) associated with important quartz content and low concentrations of illite.

A rather similar scenario was detected in some dark-colored bodies of the miscellaneous sherds (n°3, 17, 27, 35 and 36), regarding quartz and illite (very abundant/abundant),

calcite (abundant) and albite (detected). Dolomite is a predominant phase in sample n°17, there is a marked presence of chlorite in samples n°27 and 35 and sanidine is abundant in sample n°36. A common mineralogical association was noted in samples n°14 and 21, with high contents of quartz and illite, the significant presence of feldspars (albite and sanidine) and abundant chlorite. Very abundant albite, illite and sanidine and a low content of quartz were identified in sample n°23. In bodies n°33 and 34, very abundant quartz and illite, low K-feldspar content and the presence of calcite were detected.

The absence/occurrence of precise mineral phases allows the estimation of the approximate firing temperatures of the potsherds (Table 2) and the hypothesis of the pristine mineral composition of the raw materials [5,10]. Regarding pots belonging to petro-fabric 1, since quartz gradually decomposes from 800 to 1100 °C [55] and quartz and feldspars can remain in firing up to 1000 °C [15], illite is the most suitable indicator phase in order to constrain the firing temperatures of the analyzed potsherds. The illite structure begins to lose its hydroxyl group circa of 700 °C and the total breakdown of the dehydroxylated phase takes place between 850 and 950 °C [22]. As a well-preserved illite mineral points out a low temperature [15], sherds where very abundant illite (the mineral and/or the dehydroxylated phase) was detected should have been fired under 800 °C. In that the progressive decline of illite indicates increasing temperatures, approximative temperatures of 800–850 °C and 850–900 °C have been stated for the firing temperatures of the sherds, according to the abundant and noticeable illite detection, respectively. The absence of illite in sample n°10 indicates higher firing temperatures (900–950 °C). Maybe due to the relation of this pot with copper production (high contents of Cu were detected, Figure 6), especially fire-resistance pieces were required; hence, they were fired under higher heating conditions.

The firing temperatures under 850–900 °C prevented the development of high-temperature phases. In fact, only the incipient formation of ilmenite and spinel in pastes n°7 and 19 (over-fired sherds) has been detected. This would confirm that higher temperatures for these specimens were reached (900–950 °C), leading to the total dihydroxylation of illite and the early nucleation of ilmenite and spinel neo-formed phases during firing.

Regarding chlorite presence, as it decomposes between 700 and 750 °C [9], the pots where chlorite has been significantly detected (n°14, 21, 27 and 35) must have been fired below such temperatures. The abundant presence of dolomite noted in sherd n°17 may be indicative of very low firing temperatures, as dolomite decarbonation begins at about 500 °C [56]. However, as this pot was tempered (with silicate and carbonate inclusions), the approximative firing temperature (750–800 °C) has been estimated on the basis of both dolomite and illite presence. The detection of microcline instead of sanidine in sample n°27 entails other data that confirm this sherd belongs to a completely different production than those that manufactured the other dark-colored bodies with silica inclusions. Hence: (i) it is the only sherd in which a slip was probably applied on the external surfaces, (ii) it presents a unique petrological feature, corresponding to medium-grained micritic limestone fragments with irregular shapes and (iii) it was produced by using base clays with microcline as the dominant K-feldspar and that were chemically very different in terms of silicate products, clay minerals and trace element contents.

**Table 2.** Mineral assemblages detected by XRPD in the ceramic pastes and firing temperatures. Mineral abbreviations after [57]: quartz (Qz), albite (Ab), illite (Ilt), sanidine (Sa), chlorite (Chl), calcite (Cal) and dolomite (Dol). -: not detected, x: noticeable, xx: abundant, xxx: very abundant.

| | | n° | Qz | Ab | Ilt | Sa | Chl | Cal | Dol | Firing T (°C) |
|---|---|---|---|---|---|---|---|---|---|---|
| **Petro-Fabric 1. Potsherds with Silicate Inclusions** | Petro-fabric 1a | 13 | xxx | xx | xxx | x | - | - | - | <800 |
| | | 24 | xxx | x | xx | x | - | x | - | 800–850 |
| | | 26 | xxx | x | x | - | - | - | - | 850–900 |
| | | 30 | xxx | x | xx | xxx | - | - | - | 800–850 |
| | | 31 | xxx | x | xx | - | - | - | - | 800–850 |
| | | 32 | xxx | x | xx | - | - | - | - | 800–850 |
| | | 37 | xxx | xx | xxx | x | - | - | - | <800 |
| | | 39 | xxx | xx | x | - | - | - | - | 850–900 |
| | | 40 | xxx | xx | xxx | xxx | - | - | - | <800 |
| | Petro-fabric 1b | 8 | xxx | xx | x | xxx | - | - | - | 850–900 |
| | | 16 | xxx | xx | x | xx | - | - | - | 850–900 |
| | | 18 | xxx | xxx | xx | xxx | x | x | - | ≈800 |
| | | 20 | xxx | xxx | x | xx | - | x | - | 850–900 |
| | | 28 | xxx | xx | xx | x | - | - | - | 800–850 |
| | | 11 | xxx | xx | xx | x | - | - | - | 800–850 |
| | | 12 | xxx | xx | xx | x | - | - | - | 800–850 |
| | | 29 | xxx | xxx | xxx | xx | - | - | - | <800 |
| | Petro-fabric 1c. Si-rich tempering | 9 | xxx | xxx | x | xxx | - | - | - | 850–900 |
| | | 10 | xxx | x | - | - | - | x | - | 900–950 |
| | | 15 | xxx | x | xx | x | x | - | - | ≈800 |
| | | 7 * | xxx | xxx | - | xxx | - | - | - | 900–950 |
| | | 12 | xxx | xx | xx | x | x | x | x | ≈800 |
| | | 19 * | xxx | xx | - | xx | - | x | - | 900–950 |
| | | 22 | xxx | xx | xx | x | x | x | x | ≈800 |
| **Petro-Fabric 2. Calcite-Tempered Potsherds** | | 1 | xx | x | x | - | - | xxx | - | 800–850 |
| | | 5 | xx | - | x | - | - | xxx | - | 800–850 |
| | | 6 | xx | - | x | - | - | xxx | - | 800–850 |
| | | 2 | xx | - | x | - | - | xxx | - | 800–850 |
| | | 25 | xx | - | x | - | - | xxx | - | 800–850 |
| | | 38 | xx | - | x | - | - | xxx | - | 800–850 |
| **Miscellaneous Petro-Fabric 1.** | | 3 | xx | xx | xx | - | - | xx | - | 800–850 |
| | | 17 | xxx | x | xx | - | - | xx | xxx | 750–800 |
| | | 27 * | xxx | x | xxx | xx | xxx | x | - | <700 |
| | | 35 | xxx | x | xx | - | xx | xx | - | 700–750 |
| | | 36 | xxx | xx | x | xx | - | xx | - | 800–850 |
| | | 14 | xxx | xx | xxx | xx | xx | - | - | <700 |
| | | 21 | xxx | xx | xxx | xx | xx | - | - | <700 |
| | | 23 | xx | xxx | xxx | xxx | - | - | - | <800 |
| | | 33 * | xxx | x | xxx | x | - | x | - | <800 |
| | | 34 | xxx | x | xxx | - | - | xx | - | 800–850 |

* 7 and 19 (ilmenite and spinel detected), 27 (microcline instead of sanidine), 33 (noticeable anorthite instead of albite).

Even though the thermal decomposition of calcite begins circa of 750 °C and is completed up to 900 °C [58], when dealing with calcite-tempered pastes it is not appropriate to rely

on the presence of calcite to establish the firing temperatures (as seen in the pot n°17, also tempered). As a low presence of illite was detected in such sherds, firing temperatures about 850–900 °C could be estimated. However, this low presence of illite is also due to the lower clay mineral contents present in the raw materials used to produce these calcite-tempered pots. Therefore, a firing temperature between 800 and 850 °C has been stated.

In view of the rather similar mineralogical assemblages detected in sherds belonging to petro-fabric 1, a common geo-resource was probably used. This might be composed of illitic-rich (and chloritic) clays with high silica content (mainly quartz, albite and sanidine). Taking into consideration the mineralogical composition of the sand and clay fractions of the sediments of the rivers that flow through the area [23], the dark-colored pots with silicate inclusions may correspond to a local/regional pottery production. Moreover, the firing of non-calcareous illitic–chloritic clays gives rise, on the one side, from 900 °C upwards to spinel-type phases, mullite and cristobalite [59,60]. On the other, over 1000 °C to cordierite, hercynite and ilmenite in Al-rich systems [61,62] (16–21 wt% of alumina content has been detected in samples of petro-fabric 1) that can be formed in turn as a result of a reduction atmosphere [9,52,63]. The calcite-tempered pots (petro-fabric 2) were made out of illitic clays with quartz, corresponding to the calcite detected in the temper.

The statistical treatment by cluster analysis of the XRPD data (Figure 9) showed that almost all the bodies of petro-fabric 1 were grouped in clusters 1, 2, 3 and 4, according to the mineral phase abundance and therefore also to their firing temperature. Samples belonging to petro-fabric 2 fall in cluster 5, except the outlying sample n°38, related to its exceptionally high calcite content. In fact, it is the calcite-tempered pot with the highest quantity of calcite inclusions (Figure 5g,h) and with the highest lime content (CaO = 34.3 wt%). The miscellaneous dark-colored bodies are clearly isolated within clusters 1, 2, 3 and 4, and the outlying of sample n°3 is mainly because of the calcite addition to the quartz-rich illitic base clays.

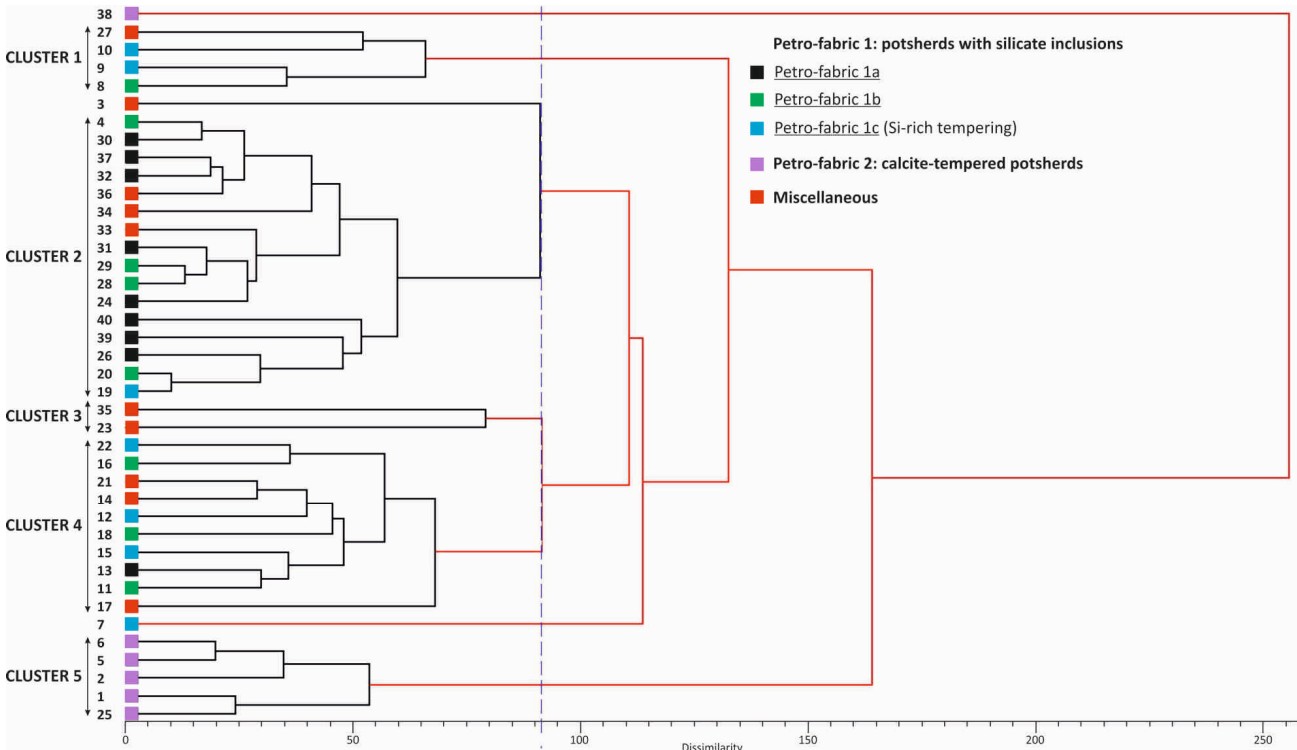

**Figure 9.** Dendrogram obtained by cluster analysis on XRPD data for the studied potsherds (according to Euclidean distance and average linkage method on position of peaks). The majority of the potsherds belonging to petro-fabric 1 were grouped in clusters 1–4 and almost all the calcite-tempered bodies fall in cluster 5.

## 4. Conclusions

The petrographic and mineralogical analysis of a set of potsherds from the site of the earliest foundry of Pre-Roman Padua (north-eastern Italy) pointed out the use of different production recipes in pottery manufacturing. Since the site is attested to have an artisanal character (for metal and pottery production) but was also a settlement, the issue of the provenance of the pottery has a fundamental implication in properly constraining the relationships of this community with regional and extra-regional coeval contexts. The analyzed potsherds correspond with two main productions:

1.  Local production from the southern Veneto territory (petro-fabric 1, dark-colored ceramic bodies). Common geo-resources were used, consisting in illitic–chloritic clays rich in quartz and alkaline feldspars. Firing regime conditions with a wide range of maximum firing temperatures (always <950 °C) and consistent with pit firing methods were adopted. For the manufacturing of the fine wares, the purification of the raw clays (delicate pieces of petro-fabric 1b) and/or the tempering with silica-rich inclusions (petro-fabric 1c) were accomplished.
2.  Regional and/or extra-regional calcite-tempered pottery (petro-fabric 2, dark- and brown-colored ceramic bodies). The pots corresponded to diverse productions, made with different illitic clays, that adopted different technological choices, mainly related to the quantity of calcite-temper addition and to the redox firing conditions.

The tempering of clay pastes with silicate and carbonate inclusions of diverse grain-size, obtained from the sieving of the base clays and grinding, entails an identifying technological choice of pottery produced during the Early Iron Age in southern Veneto. The very specific procedures adopted for the production of fine wares may point to different local/regional productions and/or the adoption of specific technological choices within the same production. The location of specific potsherds in the Questura's site may also point out the regional trade of fine wares during the Early Iron Age. The study carried out has increased the knowledge of pottery production from the Final Bronze to First Iron Ages in Padua's territory [64–66].

**Author Contributions:** E.M.P.-M., L.M., V.B. and M.V. conceived the research; E.M.P.-M. carried out the analysis, interpreted the results, wrote and supervised the manuscript; L.M. interpreted the results and reviewed the manuscript; V.B. collected the samples, provided the archaeological information and revised the manuscript; M.V. acquired funding and reviewed the manuscript. All authors have read and agreed to the published version of the manuscript.

**Funding:** This research was funded by the Project of Excellence CARIPARO Call 2021 "The earliest foundry of Pre-Roman Padua" (VIDA_ECCE22_01).

**Data Availability Statement:** No new data were created.

**Acknowledgments:** The authorization given by the Archaeological Superintendency of Veneto to sample and study the potsherds analyzed is deeply acknowledged. Special thanks are given to Michaela Ruzzante, as the typological and chronological study of the pottery from the Early Iron Age found in the Questura's site that she addressed has provided invaluable knowledge for the archaeological study carried out. The authors would like to thank the technicians of the Department of Geosciences of the University of Padua, Nicola Michelon for the photographs of the freshly-cut surfaces of the potsherds, Leonardo Tauro for the thin-sections, Marco Favero for the XRPD analysis and Daria Pasqual for the chemical analysis. The reviews and suggestions provided by the anonymous referees are gratefully appreciated.

**Conflicts of Interest:** The authors declare no conflict of interest. The funders had no role in the design of the study; in the collection, analyses, or interpretation of data; in the writing of the manuscript; or in the decision to publish the results.

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
