# Peer review of "Production Technologies and Provenance of Ceramic Materials from the Earliest Foundry of Pre-Roman Padua, NE Italy"

_heritage, doi:10.3390/heritage6030157_

Round 1

Reviewer 1 Report

Nice paper and I always love a ceramic discussion that uses multiple lines of evidence. Figures are super nice - although sometimes captions need to have more information. Overall, my main comments are:

1) Careful with verb tense - I know it is hard when talking about archaeology.  When talking about what folks did in the past, be sure to use past tense.  For example - in the abstract, lines 24-25, should read “clayey materials and of calcite inclusions as tempers might also have taken place”

2) Need a good edit of English in general.  For example - in the Introduction, line 31, should read “when documental sources are lacking”. Or line 40 should read “on the one side” and line 43 should read “On the other”. Another example, page 4, line 151, I think you mean “All vessels were wheel thrown” (not Any pot was wheeled). Also numerous places where words should be plural. 

3) Figure 4 caption reads “images were taken at the same magnification” - What magnification is that? x10? x25? Etc. 

4) Table 1 caption reads “abundant” and “very abundant”. How was this quantified? Did you use a petrography chart? Is it based on precent (for example, 25-50%) of inclusions? State in the Table caption what these terms mean (for example - xx:abundant (25-50% of inclusions), xxx: very abundant (>than 50% of inclusions).  

5) Figure 6: technically you are not plotting chemical composition, you are plotting chemical compounds. Usually what is meant by chemical composition are elements

Author Response

REVIEWER 1

Nice paper and I always love a ceramic discussion that uses multiple lines of evidence. Figures are super nice - although sometimes captions need to have more information.

Authors’ response to reviewer 1

Some information has been added at captions of Figures 5, 6 and 7.

  • Careful with verb tense - I know it is hard when talking about archaeology. When talking about what folks did in the past, be sure to use past tense.  For example - in the abstract, lines 24-25, should read “clayey materials and of calcite inclusions as tempers might also have taken place”

Answer: done.

  • Need a good edit of English in general. For example - in the Introduction, line 31, should read “when documental sources are lacking”. Or line 40 should read “on the one side” and line 43 should read “On the other”. Another example, page 4, line 151, I think you mean “All vessels were wheel thrown” (not Any pot was wheeled). Also, numerous places where words should be plural.

Answer: done. Lines 40 and 43 have been changed. According to Ruzzante [32], any pot was wheel thrown.

  • Figure 4 caption reads “images were taken at the same magnification” - What magnification is that? x10? x25? Etc.

Answer: we have changed it to “the scale bar is the same for all the images”.

  • Table 1 caption reads “abundant” and “very abundant”. How was this quantified? Did you use a petrography chart? Is it based on percent (for example, 25-50%) of inclusions? State in the Table caption what these terms mean (for example - xx: abundant (25-50% of inclusions), xxx: very abundant (>than 50% of inclusions).

Answer: we have assigned the terms according to the height/intensity of the main reflections of each mineral phase and observing jointly all the diffractogram patterns, that were compared using the same scale for the Y axis, hence the same number of counts. Therefore, these terms correspond with a semiquantitative content of each mineral phase but no with the general percent of inclusions (included at the petrographic description section), that have been quantified under the optical microscopy without using petrography chart.

  • Figure 6: technically you are not plotting chemical composition, you are plotting chemical compounds. Usually what is meant by chemical composition are elements

Answer: in Figure 6 caption, we have changed “Plotting of the chemical composition of potsherds in ternary diagrams” by “Ternary diagrams of some selected oxides or oxides ratios”.

Reviewer 2 Report

The paper was very interesting, and contain interesting results and objectives from Italy.

However, I mentioned some minor suggestions which should help the redaers for better underestanding some reactions. I have embeded all revision within the original text, upload bellow .

In general I decided to have my opinio as accepted with minor revisions.

Yours

Author Response

REVIEWER 2

The paper was very interesting, and contain interesting results and objectives from Italy. However, I mentioned some minor suggestions which should help the readers for better understanding some reactions. I have embeded all revision within the original text, upload bellow. In general, I decided to have my opinion as accepted with minor revisions.

Authors’ response to reviewer 2

The authors’ response has been included in the revised manuscript (track changes version), bellow each comment of the reviewer.

Reviewer 3 Report

Dear authors, 

You can improve the next sections for a better understanding:

For the abstract, you can write the methodology and the techniques used in this research.

For the point 2. Materials and methods

Could be improve to explain the phases of the study: global examination, like the registration, classification and formal description of the potsherds with visible imaging, drawings...

The second phase: spectroscopies,  XRF for the elemental composition and then XRD for the crystalline and amorphous phase caused by temperature...

Third phase: the petrography.

For results: you can present the aspects of the POM, then the elements identifying by XRF and for the end you can write about the crystalline phase, temperature and other results from XRD.

About the citations: there are incomplete items in the references and duplicate works by one author.

Author Response

Dear authors,

You can improve the next sections for a better understanding:

Authors’ response to reviewer 3

  • For the abstract, you can write the methodology and the techniques used in this research.

Answer: the abstract has been changed for a better understanding. Besides, we have adapted it accordingly to your suggestion. The acronym of the techniques used are entirely written at the methodological section.  

  • For the point 2. Materials and methods

Could be improve to explain the phases of the study: global examination, like the registration, classification and formal description of the potsherds with visible imaging, drawings...

The second phase: spectroscopies, XRF for the elemental composition and then XRD for the crystalline and amorphous phase caused by temperature...

Third phase: the petrography.

Answer: we have preferred to consider the first phase suggested not as a proper methodological phase of our study, as these tasks were performed by Ruzzante [32]. So, at sub-section 2.1 we have left the first sentence as it was. We have considered three methodological phases adapting your suggestion (stated at sub-section 2.2).

We want to point out that by means of XRPD analysis, the pristine, firing and/or secondary phases can be detected. In our study, the amorphous phase caused by temperature has not been determined as we have not performed qualitative analysis (by Rietveld refinement).

  • For results: you can present the aspects of the POM, then the elements identifying by XRF and for the end you can write about the crystalline phase, temperature and other results from XRD.

Answer: done.

  • About the citations: there are incomplete items in the references and duplicate works by one author.

Answer: corrected.

Reviewer 4 Report

The article describes novel results on ancient ceramics from Padua in Italy. Many results on morphology, chemical and mineralogical composition are presented. A statistical analysis (cluster analysis and PCA) and chemical analysis ternary diagrams is employed to sort the obtained results. Most images are of a good quality and the story is well presented. The literature referenced covers a wide range of papers, both old and novel ones.

-          Please re-read the whole text and try avoiding some wordy (lines 40-43, 679-685, 686-689, 689-693) and unfinished (lines 43-46, 91-94) sentences.

-          Please remove the word “instead” in line 36.

-          Improve English in lines 51-53, 89-91, 338-340, 432-433 (arrangement of words in the sentence).

-          Change “low” to “lower” in line 55.

-          Change “es” to “as” in line 52. Do a spell and grammar check throughout the text.

-          The harmfulness of calcite in clay to a large extent depends on the grain size and the content of clay minerals in the starting material. Please, add additional comment on that in the text (you can read the explanation and refer to http://dx.doi.org/10.1016/j.ceramint.2012.09.086). Please, pay attention to that some of the analyzed pottery was fired in the reduction atmosphere.

-          What is meant by “dolinas” in line 129?

-          Please define and reference US 1573 and others mentioned in Fig. 2.

-          Overfiring generally means that the material is melted so much that it starts to lose its meant shape. The reason for partial loss of the polishing maybe a mechanical damage (line 236).

-          Maybe the large pores in the inner parts of the samples 5 and 25 might be of organic material often used in the initial materials (lines 238-241). Fig. 3 could be enlarged so all the parts could be better visible. The oragnics seems to be added to cause the reducing atmosphere that is concluded to appear “at the end of firing” (line 270), which is then preserved while cooling. This reducing atmosphere for some of the vessels means different mineral transformations during firing, so discussion should be changed according to that.

-          “size-grained inclusions” is not a very good expression. Maybe grain-sized, but the exact size would be better to be defined.

-          “reduced conditions during heating and cooling” should be changed to a reducing atmosphere or environment.

-          Fig. 4 could be better organized so all the images can be better visible. Maybe it can be combined with Fig. 3 to show 3 separate images that describe all of the 3 macro-groups of vessels.

-          What is meant by a “silky” appearance? Are those surfaces smooth and glassy (items number 14, 21 and 35)?

-          Clays break down at relatively low temperature, around 500 -600 °C, so you might consider the complete break down from above around 850-900 °C (lines 433-436)? It is not clear how that would affect the “silky” appearance.

-          Change “appropriated” to “appropriate” in line 515.

-          The quality of Fig. 6 should be improved.

-          The concentration of trace elements should be briefly stated in the text or introduced in a table.

-          Try avoiding wordy sentences in the conclusion section and be more specific without so many digressions. Also, a conclusion section is not supposed to refer to the previous literature as it must be able to “stand-alone”.

Author Response

REVIEWER 4

The article describes novel results on ancient ceramics from Padua in Italy. Many results on morphology, chemical and mineralogical composition are presented. A statistical analysis (cluster analysis and PCA) and chemical analysis ternary diagrams is employed to sort the obtained results. Most images are of a good quality and the story is well presented. The literature referenced covers a wide range of papers, both old and novel ones.

Authors’ response to reviewer 4

  • Please re-read the whole text and try avoiding some wordy (lines 40-43, 679-685, 686-689, 689-693) and unfinished (lines 43-46, 91-94) sentences. Done.

  • Please remove the word “instead” in line 36. Done.

  • Improve English in lines 51-53, 89-91, 338-340, 432-433 (arrangement of words in the sentence). Done.

  • Change “low” to “lower” in line 55.

Answer: change addressed according also to suggestion of reviewer 2.

  • Change “es” to “as” in line 52. Do a spell and grammar check throughout the text. Done

  • The harmfulness of calcite in clay to a large extent depends on the grain size and the content of clay minerals in the starting material. Please, add additional comment on that in the text (you can read the explanation and refer to http://dx.doi.org/10.1016/j.ceramint.2012.09.086). Please, pay attention to that some of the analyzed pottery was fired in the reduction atmosphere.

Answer: addition done. Regarding to the second comment, see our answer in comment 10.

  • What is meant by “dolinas” in line 129?.

Answer: the correct term is doline (or sinkhole as the American term), a natural enclosed depression typically funnel-shaped found in karsts landscapes. Dolines are the most common landform in karsts areas. In any case, we have removed the term of doline from text.

  • Please define and reference US 1573 and others mentioned in Fig. 2.

Answer: US is the Italian acronym of stratigraphic unit. The correspondence between the progressive number used by authors for each sherd with denomination assigned by Ruzzante [32] that included the US where the sherd was found and the number of fragment, is shown at Figure 2. The profile with the stratigraphic units referenced is not shown at this manuscript.

  • Over-firing generally means that the material is melted so much that it starts to lose its meant shape. The reason for partial loss of the polishing maybe a mechanical damage (line 236). Ok, changed.

  • Maybe the large pores in the inner parts of the samples 5 and 25 might be of organic material often used in the initial materials (lines 238-241). Fig. 3 could be enlarged so all the parts could be better visible. The organics seems to be added to cause the reducing atmosphere that is concluded to appear “at the end of firing” (line 270), which is then preserved while cooling. This reducing atmosphere for some of the vessels means different mineral transformations during firing, so discussion should be changed according to that.

Answer: certainly, organic materials often used in the initial materials may yield large pores due to their decomposition during firing. However, in this case, the holes observed in the external surfaces of such sherds was most probably formed due to the reason suggested by Ruzzante [30]. 

The size of the fragments at Figure 3 has been enlarged.

We have re-written the paragraph where the former line 270 was, where the comment of the reviewer has been considered.

Regarding to your last comment, likewise refeed in comment 6, local reduction conditions were produced in the vicinity of the organic inclusions, so that black boundaries are observed around some large pores (which actually affect very little volumes of the ceramic body). This means that the ceramic did not underwent the same transformation along all its body. What we seen under a mineralogical viewpoint is the minerals more abundant in the ceramics: some of them are surely under the detection limit and are those that were locally formed in few microstructural sites (therefore less abundant).

  • “size-grained inclusions” is not a very good expression. Maybe grain-sized, but the exact size would be better to be defined.

Answer: in Figure 3, “diverse size-grained inclusions” has been changed by “diverse grain-size inclusions”. The sizes of the inclusions according to coarse- (500 mm-2 mm), medium- (500-250 mm) and fine- (250-63 mm) grained sand have been included at section 3.1 and conveniently changed throughout text.

  • “reduced conditions during heating and cooling” should be changed to a reducing atmosphere or environment. Done.

13) Fig. 4 could be better organized so all the images can be better visible. Maybe it can be combined with Fig. 3 to show 3 separate images that describe all of the 3 macro-groups of vessels.

Answer: we would prefer to show Figures 3 and 4 separately, as the combination of the macroscopic and petrographic descriptions would imply the change of the methodological approach. In order to achieve the better visibility of all the images of Figure 4, three sub-figures 4 (4.1, 4.2 and 4.3) have been considered.

14)  What is meant by a “silky” appearance? Are those surfaces smooth and glassy (items number 14, 21 and 35)?

       Answer: as this appearance is better perceived with the touch than with the view, we firstly though of using the term “silk touch”. As this term can be somehow confusing for the reader, we thought about the term “glassy surface” or “vitreous appearance”. However, these terms might likewise confuse the reader as, dealing with fired clays they directly refer to the sintering and vitrification processes. Moreover, an important chlorite content was detected, so the terms “glassy surface” or “vitreous appearance” were in turn completely wrong. From the cut of the sherds in order to describe the freshly cut surfaces, this special appearance, observed in just three sherds, (corresponding to very delicate pots) caught our attention, even more when a significant content of chlorite was detected by XRPD just in these three potsherds. As these surfaces display a blurred appearance, silky has been changed by blurred. Certainly, such freshly-cut surfaces are very smooth, but the external surfaces of some pots were smoothed and our intention is to highlight this particular feature, that may be useful to identify specific pottery productions.

15) Clays break down at relatively low temperature, around 500 -600 °C, so you might consider the complete breakdown from above around 850-900 °C (lines 433-436)? It is not clear how that would affect the “silky” appearance.

Answer: certainly, chlorite breaks down at low temperatures and it decomposes between 700-750 °C. However, illite structure begins to lose its hydroxyl group circa of 700 °C and the total breakdown of the dehydroxylated phase take places between 850-950 °C, as stated in the references provided. In order to a better clarification, at the end of section 3.2, in the revised version we have written (lines 402-404): the purification of the raw clays was also carried out to produce sherds n°14, 21 and 35, that were in turn fired at low temperatures (below the breakdown of phyllosilicate, as denotes the optically active groundmass)”. The possible relation of the silky/blurry appearance noted on freshly cut surfaces with the adoption of both technological choices (purification of raw clays and low firing temperatures) has been removed.

16) Change “appropriated” to “appropriate” in line 515. Done.

17) The quality of Fig. 6 should be improved. Done.

18) The concentration of trace elements should be briefly stated in the text or introduced in a table. Done.

19) Try avoiding wordy sentences in the conclusion section and be more specific without so many digressions. Also, a conclusion section is not supposed to refer to the previous literature as it must be able to “stand-alone”. Done. We would like to maintain, if possible, last sentence as it was.

Round 2

Reviewer 3 Report

Congrats for all your work. I don't have any suggestion.